# Chemical Profile and Promising Applications of *Cucurbita pepo* L. Flowers

**DOI:** 10.3390/antiox13121476

**Published:** 2024-11-30

**Authors:** Ritamaria Di Lorenzo, Luigi Castaldo, Raffaele Sessa, Lucia Ricci, Eleonora Vardaro, Luana Izzo, Michela Grosso, Alberto Ritieni, Sonia Laneri

**Affiliations:** 1Department of Pharmacy, University of Naples Federico II, Via Domenico Montesano, 49, 80131 Naples, Italy; ritamaria.dilorenzo@unina.it (R.D.L.); luigi.castaldo2@unina.it (L.C.); lucia.ricci@unina.it (L.R.); eleonora.vardaro@unina.it (E.V.); slaneri@unina.it (S.L.); 2Department of Molecular Medicine and Medical Biotechnology, University of Naples Federico II, Via Pansini 5, 80131 Naples, Italy; raffaele.sessa@unina.it (R.S.); michela.grosso@unina.it (M.G.)

**Keywords:** *Cucurbita pepo*, polyphenols, carotenoids, photo protection, sustainability, Q-Exactive

## Abstract

Although edible flowers have been historically principally used due to their visual appeal and smell, the world is discovering their value as innovative and natural sources of bioactive compounds. *Cucurbita pepo* L. (CpL), a plant from the Cucurbitaceae family, is widely cultivated for its edible fruits and flowers, which are rich in polyphenols and carotenoids—compounds known for their potent antioxidant and anti-inflammatory properties. Despite their potential, the use of CpL flowers for skin-related applications remains underexplored. This study aimed to comprehensively analyze CpL flower extract (CpLfe), focusing on its polyphenolic and carotenoid content using, for the first time, advanced UHPLC-Q-Orbitrap HRMS and HPLC-DAD analysis. CpLfe highlighted remarkable antioxidant activity according to the DPPH, ABTS, and FRAP tests. CpLfe showed significantly reduced intracellular ROS in HaCaT (23%, *p* < 0.05) and protected against UVB-induced damage by lowering MMP-1 expression. CpLfe also upregulated genes crucial for skin hydration (AQP3) and barrier function (CerS2, CerS4, and CerS6). A placebo-controlled, randomized clinical trial further validated CpLfe efficacy, demonstrating marked improvements in moisture retention, wrinkle reduction, and collagen production in women aged 35–55. These findings suggested that CpL flowers could be a source of bioactive compounds recovered from edible flowers able to improve the major skin aging and photoaging features.

## 1. Introduction

Over the past few decades, edible flowers have gained more interest due to their appealing taste, color, pleasant aroma, and concentration of active ingredients. According to scientific evidence, edible flowers contain substantial quantities of minerals, vitamins, and antioxidant molecules that could justify their exploitation as innovative ingredients [1,2].

*Cucurbita pepo* L. (CpL) is a plant that belongs to the Cucurbitaceae family native to Mexico [3]. Currently, it is widely cultivated all over the world for its edible fruits. Along with the fruits, the flowers of CpL are also consumed in dish decorations, salads, and desserts due to their delicate texture, intense color, and slightly sweet flavor [4]. Previous studies have suggested that CpL flowers contain considerable levels of active compounds such as polyphenols and carotenoids [5]. Carotenoids are naturally occurring chemical molecules with yellow, orange, and red colors well-known to exert potent antioxidant activity [6]. Scientific data suggested that these compounds may serve as a natural alternative for synthetic chemicals, due to their respect for the environment and the protection of biodiversity.

On the other hand, polyphenols are a group of active compounds found in many plant-based foods, including vegetables, nuts, tea leaves, and fruits, among others [7,8]. In general, these above-mentioned compounds are well-known to exert potent antioxidant and anti-inflammatory activity [9,10,11,12,13]. A growing amount of data indicates that polyphenols could exert several bioactive effects on the human skin, representing an excellent functional ingredient for cosmetic purposes [14,15,16,17,18]. The use of active compounds deriving from plants or other vegetable sources as promising ingredients to support skin health and well-being has gained significant attention due to the growing environmental and health awareness. Therefore, employing plant-based extracts and their phyto-components as active ingredients represents a modern, eco-sustainable way in the formulation of novel cosmetics or cosmeceuticals [19,20,21,22].

Some preliminary results suggested that CpL flowers could be a source of bioactive compounds recovered from edible flowers [23,24]. In fact, it was reported that applying active antioxidants topically can help the skin take on oxidative conditions and provide a long-term defense against photoaging [14,25,26]. Data showed that their extracts enhanced skin barrier function by promoting the production of protein kinase C, p38, and ERK 1/2, acting as skin moisturization upregulating epidermal involucrin, representing a valid source of active ingredients to be used in cosmetics with several claims, including skin conditioning agents, hydration, and anti-aging [27,28].

The aging process is physiological and natural; however, harmful external factors such as pollution and UV radiation can accelerate it. These conditions promote the generation of free radicals as a result of an incomplete reduction of oxygen molecules [29,30]. Free radicals easily involve cell components in chemical reactions, leading to lipid oxidation, protein structural conversion, and damage to nucleic acid structures [31]. Lipid peroxidation affects both cell components and membranes, determining their increased permeability [32,33]. Skin is constantly exposed to harmful exogenous effects, by virtue of its barrier function for our organism [30,34,35,36]. Although skin has its natural defense mechanisms, it becomes vulnerable to ROS when they are produced in excessive amounts [37,38,39]. ROS affects the epidermis and dermis, inducing complex cellular responses with activation of pathways associated with the expression of telomerase genes, inflammation, angiogenic and anti-apoptotic factors, and cellular proliferation [40]. Moreover, ROS contributes to melanogenesis alteration, leading to the appearance of melasma lesions [35]. ROS may also induce the expression of serine proteases and matrix metalloproteinases (MMPs), responsible for collagen degradation, accelerating skin aging [41]. Hence the need to counteract the propagation of free radicals through scattering processes, which, notoriously, the botanicals antioxidant molecules are able to perform [42,43,44].

In recent years, topical products have incorporated functional ingredients from food sources such as cherries, pomegranates, apples, and tomatoes to increase their properties [45,46,47,48,49]. Despite being a potential source of natural compounds for cosmetics, there is a shortage in the literature on the use of CpL flowers as topical cosmetic agents. Recent analysis using high-pressure liquid chromatography (HPLC) found significant concentrations of flavonoids, including rutin and quercetin glycoside, in the ethanol extracts of the CpL flowers [50,51]. However, the LC technique has advanced in the last decade with the advent of ultra-HPLC, resulting in shorter analysis time, improved peak efficiency, and increased resolution. Moreover, high-resolution mass spectrometers (HRMS), such as Q-Orbitrap, coupled with UHPLC, represent an effective method for detecting and quantifying natural compounds, including polyphenols in plant-based materials. This powerful tool offers high specificity and sensitivity, providing accurate mass measurement-based quantification.

Hence, the aim of this study was to (i) provide a comprehensive analysis of the polyphenolic and carotenoid composition contained in the propylene glycol extract of CpL flowers; (ii) estimate the antioxidant potential and total polyphenol content (TPC) of CpLfe through in vitro assays; (iii) investigate the ROS scavenging action; (iv) assess photoprotection; (v) evaluate the stimulation of skin wellness markers using pre-clinical cellular screenings on HaCaT cells; and (vi) determine skin tolerability and efficacy on skin barrier recovery, hydration, collagen levels, wrinkles, and dark spot lesions through a placebo-controlled, randomized clinical trial. This multifaceted approach aims to justify the use of CpL flowers beyond their conventional application as an active ingredient able to boost skin health and well-being.

## 2. Materials and Methods

### 2.1. Chemicals and Reagents

Methanol, ethanol, formic acid, water (UHPLC grade), *n*-hexane, chloroform, dichloromethane, propylene glycol, polyphenolic, and carotenoid standards (purity > 98%) were purchased from Merck KGaA (Darmstadt, Germany).

### 2.2. Sampling

CpL plants were grown in different fields located in the Lazio region (central Italy). All edible flowers were manually harvested in July 2022. CpL flower samples were quickly washed with cold running water, frozen, and lyophilized using freeze-drying. The samples were ground into a powder through a laboratory mill. The obtained samples were kept until the analysis at −80 °C.

### 2.3. Cucurbita pepo L. Flowers Extract Preparation

The polyphenolic compounds from CpL flowers were extracted following the procedure developed by Frankič et al. [52]. In particular, 250 mg of powdered samples was combined with 5 mL of propylene glycol and sonicated for 10 min. Afterward, the sample was vortexed for 3 min and then centrifuged for 3 min at 5000× *g*. The supernatants were filtrated and stored at −80 °C. The obtained extract was used to prepare the active emulgel after the chemical characterization.

### 2.4. UHPLC and Orbitrap HRMS Analysis

Chromatographic separation was performed by using a UHPLC device (Dionex UltiMate 3000, Thermo Fisher Scientific, Waltham, MA, USA) equipped with a degassing device, a quaternary pump, and an autosampler. For chromatographic separation, a thermostated column set at 25 °C (Kinetex F5, Phenomenex, Torrance, USA; 50 × 2.1 mm, 1.7 µm) was employed. Water and methanol, both of which contained 0.1% FA, were the eluent phases. The gradient elution was started with 100% A for 1 min, followed by 2 min of 20% A. The gradient once more decreased to 0% A in three minutes. The gradient then underwent another shift and returned in two minutes to the initial 100% A.

For detection, a negative mode Q-Exactive Orbitrap mass spectrometer (Thermo Fisher Scientific, Waltham, MA, USA) was employed in full MS mode. The following settings were applied: spray voltage 3.5 KV, auxiliary gas heater temperature 350 °C, capillary temperature 320 °C, sweep gas flow rate 0, S-lens RF level 60, sweep gas flow rate 0, scan range 80–1000 *m*/*z*, maximum injection time 200 ms, auxiliary gas 3, AGC target 1 × 10^6^, sheath gas flow rate 18, resolution power of 70,000 FWHM, and microscan 1. The data was processed using Xcalibur software 3.1.66.19. (Thermo Fisher Scientific, Waltham, MA, USA).

### 2.5. Carotenoid Extraction and Determination

The carotenoid extraction was based on the protocol described by [53]. In short, 1 g of the sample was mixed with 6 mL of 0.1% BHT in ethanol. The mixture was then vortexed for 1 min and heated in a water bath for 5 min at 85 °C. Next, 120 μL of 80% aqueous KOH was added, followed by another vortex for 1 min and a 10 min saponification step. The samples were then cooled before adding 3 mL of hexane and 3 mL of water. The mixture was centrifuged for 5 min at 4900× *g*. The hexane layer was collected, and the extraction was repeated twice more. The combined supernatants were dried using nitrogen, re-suspended in 1 mL of chloroform, and filtered with 0.2 μm nylon filters prior to analysis.

The profile of carotenoids was performed by using a Jasco HPLC Model 2000 Plus Series (Jasco, Cremella, Italy). A Gemini C18 column (250 mm × 4.6 mm, 5 m, Phenomenex, Castel Maggiore, Italy) was used for the chromatographic separation. Elution was carried out with acetonitrile (phase A) and a combination of dichloromethane, n-hexane, and ethanol in a 1:1:1 ratio (phase B). A starting concentration of 18% of mobile phase B was used in the gradient elution program, which was then gradually increased to 24% in 8 min, 42% over 4 min, and 61% over 6 min. The gradient was then reduced to 18% in 4 min, followed by an additional 5 min for column re-equilibration. The run time was 27 min, the flow rate was 1 mL/min, and the volume injection was 20 µL. The absorbance of lutein, lycopene, and b-carotene was measured at 450 nm.

### 2.6. Total Phenolic Content

The total phenolic content (TPC) was evaluated using the protocol proposed by Castaldo et al. [41]. Briefly, 125 µL of the diluted sample, 125 µL of the Folin–Ciocolteu reagent, and 1.5 mL of deionized water were mixed. Afterward, 1.25 mL of sodium carbonate solution (7.5%) was added to the mixture. After 60 min, the absorbance was observed at 760 nm. The data was displayed in milligrams of acid gallic equivalent (GAE) per gram of sample.

### 2.7. Antioxidant Activity

CpL flowers were tested for their antioxidant properties using the DPPH, FRAP, and ABTS methods. The data were displayed as mmol Trolox per kilogram of dry weight sample.

#### 2.7.1. DPPH Assay

The DPPH assay was conducted according to the procedure previously reported [9]. In short, MeOH was used to dilute the DPPH standard until the absorbance reached 0.900 ± 0.02 at 517 nm. Following that, 0.2 mL of the samples were added to 1 mL of DPPH working solution (WS). After waiting for 10 min, the absorbance at 517 nm was finally observed.

#### 2.7.2. ABTS Assay

The ABTS test was carried out in accordance with the methodology previously reported [7]. In short, 44 µL of potassium persulfate (2.45 mM) and 2.5 mL of aqueous ABTS (7 mM) were mixed. After 16 h of room temperature incubation, the ABTS mixture was diluted with ethanol until the absorbance reached 0.700 ± 0.02 at 734 nm. Finally, 100 μL of the sample were mixed with 1 mL of ABTS WS. After 2.5 min, the absorbance was measured at 734 nm.

#### 2.7.3. FRAP Assay

The FRAP test was carried out following the procedure previously reported [54]. In brief, 1.25 mL of FeCl_3_ solution (20 mM), 12.5 mL of acetate buffer (0.3 M, pH 3.6), and 1.25 mL of TPTZ solution (10 mM) were mixed. After that, 2.85 mL of FRAP reagent was added to 150 µL of samples. The absorbance measurements at 593 nm were recorded immediately after 4 min.

### 2.8. Cell Culture

The human keratinocytes HaCaT cells from the American Type Culture Collection (ATCC, Manassas, VA, USA) were cultured in Dulbecco’s Modified Eagle’s medium high glucose (DMEM), containing 10% heat-inactivated fetal bovine serum (FBS; Sigma Aldrich, St. Louis, MO, USA) and 1% GlutaMAX-I (Invitrogen, Thermo Fisher Scientific, Waltham, MA, USA). The cell cultures were maintained, under appropriate conditions, at 37 °C in a humidified 5% CO_2_^−^ incubator and kept at 60–70% confluency. To avoid mycoplasma contamination, cells were routinely checked with the PCR Mycoplasma Test Kit (AppliChem A3744, Darmstadt, Germany).

#### 2.8.1. Cell Treatments

HaCaT cells were seeded in 96-well plates (1 × 10^4^ cells/well) and cultured for 24 h before treating the cells with each concentration (50, 100, 250, 500, 750, or 1000 μg/mL) of CpLfe extract. The cells were incubated for 24 h and 48 h under the same conditions for MTT and H2DCF-DA assays. Appropriate vehicle controls were included in each experiment by treating the cells with the same amount of polyethylene glycol (0.2–1%, *v*/*v*).

#### 2.8.2. Intracellular ROS Detection

Intracellular ROS were detected using the fluorescent probe H2DCF-DA (2′7′-dichlorodihydrofluorescein diacetate) in a spectrofluorometric assay [55,56,57,58]. HaCaT cells were plated at a density of 1 × 10^5^ cells/mL in 100 µL of cell suspension per well on black 96-well plates. The CpLfe was diluted in serum-free medium and added to the culture medium at final concentrations of 100, 250, and 500 μg/mL. Cells were then incubated for 24 h. For each experiment, a positive control was included by treating cells with 300 µM hydrogen peroxide (H_2_O_2_), followed by incubation with H2DCF-DA as previously described [59]. After treatment, cells were washed twice with Dulbecco’s Phosphate Buffered Saline (DPBS) and then exposed to 10 µM H2DCF-DA, diluted in Hank’s Balanced Salt Solution (HBSS), for 20 min at 37 °C in the dark. After staining, the extracellular dye was removed, and the cells were washed twice with 1× DPBS. Fluorescence intensity was measured with a Synergy H1 Hybrid Multi-Mode microplate reader (BioTek) at excitation/emission wavelengths of 485/538 nm, and values were expressed as relative percentages compared to the control group using the following formula: (fluorescence intensity of treated cell group/fluorescence intensity of control group) × 100.

#### 2.8.3. UVB Irradiation

HaCaT cells were grown in 96- or 6-well plates under starvation conditions for 72 h and then treated with different concentrations of the CpLfe (100, 250, and 500 μg/mL) diluted in serum-free medium. After 24 h cells were then washed twice with phosphate buffered saline 1× DPBS and exposed to UVB irradiation (30 mJ/cm^2^) as previously described [60,61,62] with a UV crosslinker AH (115 V/230 V) equipped with 306 nm UVB bulbs (Boekel Scientific Inc., Feasterville, PA, USA).

After removal of PBS, serum-free DMEM was added to the cells, which were incubated for a further 24 h and then used for MTT assay and RNA extraction.

#### 2.8.4. Analysis of Cell Viability

Cell viability was evaluated by MTT assay performed as previously described [61]. Briefly, 10 µL of MTT labeling reagent (Cell Proliferation Kit I; Roche, Mannheim, Germany) was added to the cell culture, and the plate was incubated for a further 4 h at 37 °C in the humidified 5% CO_2_^−^ incubator. One hundred µL of detergent solubilization buffer 1× (10% SDS in 0.01 M HCl) was added to each well to dissolve the MTT insoluble formazan crystals according to the manufacturer’s instructions. Optical density was measured at 570/690 nm with a Synergy H1 Hybrid Multi-Mode Microplate Reader (BioTek, Winooski, VT, USA). The percentage of cell viability was calculated as follows: (absorbance of the experimental group/absorbance of the control group) × 100.

#### 2.8.5. Real-Time PCR Analysis

Total RNA was extracted from HaCaT cells with the QIAzol reagent (Qiagen, GmbH, Hilden, Germany) according to the manufacturer’s protocol. To evaluate the gene expression levels of aquaporin-3 (AQP-3), ceramide sintase 2 (CerS2), ceramide sintase 4 (CerS4), ceramide sintase 6 (CerS6), and metalloproteinase-1 (MMP-1), a quantitative real-time PCR analysis was performed. Five hundred ng of RNA were reverse transcribed using the iScript Reverse Transcription Supermix for RT-qPCR (Bio-Rad, Berkeley, CA, USA) in a final volume of 20 µL, according to the manufacturer’s instructions. The mixture was incubated at 42 °C for 3 min and then at 95 °C for 3 min and subsequently used for real-time RT-PCR procedures on a CFX96 Real-Time System (Bio-Rad Laboratories, Hercules, CA, USA). Primers for quantitative real-time PCR analysis are reported in Table 1. GAPDH mRNA was used as an endogenous control.

Each real-time PCR was carried out in triplicate in a 20 μL reaction mix that included 6.6 μL of cDNA (1/2 volume of RT-PCR product), 0.38 μL of a 20 μM primer mix, and 10 μL of 2× SsoAdvanced Universal SYBR Green Supermix (Bio-Rad Laboratories). Initial denaturation at 98 °C for 30 s was followed by 40 cycles (95 °C for 15 s; 60 °C for 30 s) under the cycling conditions. As previously reported [63], the calibration curve was performed to evaluate the efficiency of PCR reaction. The CFX Opus 96 Real-Time PCR System (Bio-Rad Laboratories) was used to conduct real-time PCR experiments, and CT values were obtained from automated threshold analysis. The CFX Manager 3.0 software (Bio-Rad Laboratories GmbH, Munich, Germany) was utilized to analyze the data in accordance with the manufacturer’s recommendations, and the relative quantification in gene expression was determined using the ΔΔCT method.

### 2.9. Clinical Trial

The skin effects of the CpLfe were evaluated through a double-blind, placebo-controlled, parallel-arm, randomized clinical study. Healthy Italian subjects (age 35–55; *n* = 40) were enrolled in a 4-week study. Before participating, each subject signed a written informed consent that contained the aim and the type of the study, the list of the cosmetic-grade ingredients employed for the preparation of the topical formulation, the rules to be followed, and any known or potential adverse reactions that might result from using the test products. The study protocol was reviewed and approved by qualified clinical, toxicology, and regulatory personnel of the university, and by corresponding personnel at the laboratory site. Due to the cosmetic nature of the study, a formal review of the ethics committee was not performed. The study was performed from October 2022 to January 2023, and it was completed with all enrolled subjects. The study protocol (unique protocol ID: EACpLfe22S01) was reviewed by the Institutional Review Board of the University IRB-DipFarm (ID No. PRO_CCT22S01) which issues opinions on compliance with ethical principles for drug-free studies using minimally/non-invasive methods, and it was registered on clinicaltrials.gov (ID: NCT06674005). 

#### 2.9.1. Inclusion Conditions

All subjects were selected based on the presence of visible wrinkles and skin laxity. Enrolled subjects underwent a 1-week washout period, in which they were instructed to stop using their usual cosmetic products, whereas they were permitted to continue their normal facial treatment regimen (e.g., facial cleanser, make-up removal, eye and lip make-up, and foundation) that did not contain ingredients with anti-aging or lightening action. Once the washout period elapsed, subjects were supplied with test products (placebo and the same formulation added with CpLfe).

#### 2.9.2. Test Samples

The ingredient list of the emulgel containing CpLfe is reported in Table 2. The placebo formulation contained all the components without CpLfe (Table 2). Cosmetic-grade ingredients were pursued by ACEF Spa (Fiorenzuola D’Arda, Italy). The emulgels preparation basically included three steps: First, the preparation of oil in water emulsion, followed by the gel base formulation step, and finally the addition of the emulsion to the gel in continuous stirring to form the emulgel. In detail, for the preparation of the emulsion, the aqueous phase (W) is prepared by taking the purified water with the hydrophilic components heated up to 69 ± 1 °C while ensuring the oil phase (O) heats up to the hydrophobic ingredients melting point (approx. 69 ± 1 °C), then shaking vigorously the two phases with a Silverson L5M-A Laboratory Mixer (SBL, Shanghai, China). Subsequently, the emulsion was cooled in an ice bath up to 30 ± 2 °C.

The gel phase is prepared by dispersing the gelling agent in purified water with constant stirring at a moderate speed, and then the pH is regulated up to 5.3 with NaOH 20% solution. Finally, the emulsion is added to the gel base in a ratio of 1:1 to obtain the emulgel. After 24 h, the physical appearance, pH, and viscosity of both test formulations (CpLfe and placebo) were checked, employing the Crison GPL 20 pH-Meter (Crison, Barcelona, Spain) and the ViscoBasic Plus Rheometer (Fungilab, Barcelona, Spain). Afterward, test samples were packaged in 50 mL blind-coded pump jars labeled by the cosmetic formulator with the volunteer’s ID identification number and the designation of “AB” or “AC” to differentiate the placebo from the active formulations. The specifically assigned designation was only known by the formulator, while all other investigators were blinded to this information.

#### 2.9.3. Instructions of Use

Panelists had to apply the assigned emulgel twice daily (morning and evening) for 4 weeks. The application amount had to be approximately 0.3 g of product (two pumps) on a dry or slightly moistened face. It had to be massaged in, avoiding direct contact with the eyes. Upon application, in case of individual hypersensitivity, panelists were instructed to discontinue treatment. It was not to be used on irritated or chapped skin. For the evening application, panelists were instructed to wait for at least 1 h before bedtime.

#### 2.9.4. Skin Condition Analysis

Skin wellness was evaluated by measuring hydration and transepidermal water loss (TEWL) using the Corneometer^®^ CM 825 and Tewameter^®^ TM Hex (Courage + Khazaka electronic GmbH), respectively. The Corneometer CM 825 measures skin hydration by assessing capacitance changes in a dielectric medium, which reflect variations in the skin’s surface hydration. The Tewameter^®^ TM Hex, on the other hand, measures TEWL through an open-chamber method. The probe quantifies the density gradient of water evaporation from the skin (Δc), which is directly proportional to the TEWL.

Digital images of all subjects were captured at baseline and at weeks 2 and 4 with VISIA^®^ 7th (Canfield Scientific Inc., Parsippany, NJ 07054 USA). The skin imaging system exploits IntelliFlash^®^, cross-polarized light to record and measure skin furrows, folds, and wrinkles. Before the image was captured, subjects were equilibrated in a controlled temperature room (22 ± 2 °C) for 30 min. The hair of the participants was tied up, and their clothing was covered with black cloth. The images were taken by the same operator using the same imaging equipment under the same conditions (lighting, distance, head position, etc.) at all points of all time. Accurate subjects’ repositioning was obtained by comparing the live image with the ghost-baseline digitally stored photo. Computer analysis of the digital images allowed quantification of the main facial expressive wrinkles (forehead and frown lines, nasolabial folds, and crow’s feet).

Facial ultrasound was performed using the Dermascan C (Cortex Technology Aps, Aalborg, Denmark) to evaluate the collagen index, based on the reflection of ultrasound by collagen fibers. This method provided insight into the structural organization of the dermis both before and after treatment, helping to confirm the ability of CpLfe to firm and densify the skin.

Lastly, facial skin color was measured using a Skin-Colorimeter^®^ CL 400 (Courage + Khazaka electronic GmbH). The device emits white LED light to illuminate the skin uniformly, and the sensor captures the reflected and scattered light, including that from deeper skin layers. The raw data are then corrected using a color matrix to align with standard values. Specifically, skin luminosity was assessed through the L* parameter in the CIELAB color space, which indicates luminosity on a scale from black (0) to white (100), with higher L* values corresponding to greater brightness.

### 2.10. Statistical Analysis

In the in vitro assays, the mean values and standard deviation (SD) of at least three different experiments were used to display the results. To assess the significance of the differences among the averages, Tukey’s test was performed (*p* value ≤ 0.05). Data processing was carried out using Stata 12 software (StataCorp LP, College Station, TX, USA). Statistical differences between untreated control and treated cells were expressed as mean ± SD and calculated using one-way ANOVA, followed by Dunnett’s multiple comparisons test (post hoc test) for more than 2 experimental groups. The level of significance was set at * *p*-value ≤ 0.05, ** *p*-value ≤ 0.0001 versus each respective negative control (vehicle), and at # *p*-value ≤ 0.05, ## *p*-value ≤ 0.0001 versus untreated cells. Cell experiments were repeated in triplicate at least three times. For clinical efficacy, skin variables were analyzed statistically using SPSS software 15.0 for Windows (SPSS Science, Chicago, IL, USA) through a Student’s *t*-test for intra-group analysis and ANOVA for inter-group differences. A *p* value < 0.05 was considered significant.

## 3. Results

### 3.1. Identification of Polyphenolic Compounds in Cucurbita pepo L. Flower

The identification of polyphenolic compounds in CpL extracts obtained using a simple propylene glycol procedure was evaluated using a UHPLC-Q-Orbitrap HRMS analysis. The present work investigated a total of 24 different analytes. Data for mass parameters such as adduction, measured and theoretical mass, accuracy, sensitivity, and retention time (RT) are displayed in Table 3. Experiments were carried out in negative ESI- mode, and full-scan HRMS was used to monitor the results. The structural isomers catechin and epicatechin (*m*/*z* 289.07175), luteolin and kaempferol (*m*/*z* 285.04046), and genistein and apigenin (*m*/*z* 269.04554) were identified by comparing the RTs of the real standards with RTs of the peaks and data obtained from the literature.

### 3.2. Quantification of Polyphenolic Compounds in Cucurbita pepo L. Flowers

Quantification of individual polyphenols in the CpL flower using UHPLC-Q-Orbitrap HRMS analysis was performed. The quantitative analysis for all studied compounds was performed using calibration curves created in triplicate at eight concentration levels (0.039–5 µg/kg). Regression coefficients obtained for each calibration curve were higher than 0.990. Table 3 displays the findings here obtained, expressed as mean content in mg per kg of extract and standard deviation (±SD) detected in the CpL flowers. The total amount of polyphenols present in the CpLfe was quantified at a concentration up to 2906.6 mg/kg of dry weight (DW) extract. Moreover, the most abundant polyphenolic compound found in the extracts of CpL flower was rutin, which accounted for 81.2% of total polyphenols present in the extracts. Apart from rutin, some important polyphenols were quantified in the assayed extracts, such as quercetin 3-galactoside, kaempferol 3-glucoside, isorhamnetin 3-rutinoside, myricetin, quercetin, and kaempferol, as shown in Table 4. Concerning other compounds, both naringin and apigenin were detected in concentration levels < LOQ in the assayed extracts.

### 3.3. Quantification of Carotenoids in Cucurbita pepo L. Flowers

The amount of carotenoids in CpL flowers was quantitatively measured using HPLC-DAD analysis. The quantitative analyses for all studied compounds, such as lutein, β-carotene, and lycopene, were performed using calibration curves created with real standards. The regression coefficients that were achieved for each calibration curve exceeded 0.990. The results of the study show that the CpL flowers contained 407 mg of lutein per kg of sample, 83 mg of β-carotene per kg of sample, and 513 mg of lycopene per kg of sample. The data are summarized in Table 5. These findings indicate that CpL flowers are a rich source of carotenoids, qualifying them as suitable ingredients for supporting skin health and well-being.

### 3.4. Antioxidant Capacity and Total Phenolic Content in Cucurbita pepo L. Flowers

The antioxidant properties of CpL flower extract were evaluated using three distinct tests, specifically DPPH, ABTS, and FRAP. The results are summarized in Table 6. The obtained data were 10.7, 12.4, and 77.5 mmol Trolox per kg of DW extracts for DPPH, ABTS, and FRAP, respectively. Moreover, the Folin-Ciocâlteu assay was also used to detect the TPC levels in the flowers of CpL. The results are displayed in Table 6. The findings highlighted a TPC value of up to 534.2 mg GAE/100 g for the assayed extract. Furthermore, the TPC data measured in the assayed extracts showed a positive correlation with the corresponding ABTS, DPPH, and FRAP data, as displayed in Appendix A.

### 3.5. Intracellular ROS Levels in HaCaT Cells

The effect of CpLfe on the intracellular ROS levels was examined by a fluorometric test using the specific dye H2DCF-DA in HaCaT cells. Cells were treated with 100, 250, and 500 μg/mL of CpLfe for 24 h, chosen because they resulted in non-toxic concentrations in preliminary MTT assays (Appendix A). Cells treated with H_2_O_2_ (300 μM) were used as a positive control. Results (Figure 1) showed that the relative fluorescence intensity significantly decreased in cells treated with 250 and 500 μg/mL CpLfe concentrations compared to untreated cells and vehicle control (polyethylene glycol). More in detail, fluorescence intensity signals were reduced by 23% and 33% after CpLfe treatment with 250 and 500 μg/mL, respectively. These results correlate with the antioxidant activity and total phenolic content previously found in CpL flower extract.

### 3.6. Protective Effects of CpL Flower Extract on UVB-Induced Cell Damage

The protective effect of CpLfe against UVB exposure was evaluated by MTT assay on cells treated with different concentrations (100, 250, and 500 μg/mL) for 24 h.

Cells were irradiated with UVB at 30 mJ/cm^2^, as previously reported [64]. MTT assay was performed 24 h after UVB irradiation. In the absence of CpLfe treatment, UVB-exposed cells showed a cell survival rate of 65% compared to non-irradiated cells. Conversely, a dose-dependent protective effect was shown in cells pre-treated with CpLfe. In fact, a higher cell viability rate was detected even at the lower tested dose (84%), which was almost completely comparable to the non-irradiated cell control in cells exposed to doses of 250 and 500 μg/mL (Figure 2a). We also evaluated the effects of UVB treatment by examining the expression levels of MMP-1, a matrix metalloproteinase whose levels are increased in the photoaging process [65,66]. Therefore, we examined the mRNA expression of MMP-1 in HaCaT cells irradiated with UVB at 30 mJ/cm^2^ and harvested after 24 h. Real-time PCR data revealed that MMP-1 expression significantly increased by approximately 30% in UVB-exposed cells as compared to non-UVB-irradiated cells. In contrast, pre-treatment of HaCaT cells with CpLfe resulted in a significant decrease of MMP-1 transcript as compared to the vehicle UVB cells. These protective effects were statistically significant in cells exposed to 250 and 500 μg/mL doses (Figure 2b). As a whole, these results provide evidence of the efficacy of the UVB treatment and the inhibitory activity of CpLfe on the UVB-induced expression of MMP-1, thus indicating a molecular mechanism through which CpLfe can exert a protective effect against UVB in HaCaT cells.

### 3.7. Effect of CpLfe on Aquaporin-3 (AQP-3) and Ceramide Synthase (CerS2, CerS4, and CerS6) Expression Levels

To evaluate the hydration properties of CpLfe, we determined the mRNA levels of AQP3 by real-time PCR analysis [67,68]. After 24 h exposure to the 250 µg/mL dose, CpLfe-treated cells showed significantly increased levels of AQP-3 compared to untreated and vehicle controls (Figure 3a).

We also evaluated the effect of CpLfe on the expression levels of ceramide synthases. As shown in Figure 3, mRNA expression levels of CerS2, CerS4, and CerS6 resulted in significant increases in cells treated for 24 h with CpLfe (250 μg/mL) (Figure 3b–d). Collectively, these findings indicate that CpLfe could play a role in the restoration of human skin barrier function [69,70].

### 3.8. Physicochemical Characterization of Formulations

After 24 h, the physical appearance, pH, and viscosity of both test formulations (CpLfe and placebo) were checked, as shown in Table 7.

### 3.9. Skin Wellness Assessment

Skin wellness was evaluated by monitoring TEWL and hydration levels (Table 8). After two weeks, the emulgel containing CpLfe reduced TEWL by 7.5% (*p* < 0.001), with a further reduction of 10.2% (*p* < 0.001) after four weeks. In contrast, the placebo showed no significant effect on TEWL at either time point. Additionally, the CpLfe emulgel significantly enhanced skin hydration, with increases of 13.7% (*p* < 0.001) and 15.0% (*p* < 0.001) after 2 and 4 weeks of treatment, respectively. In comparison, the placebo demonstrated only modest improvements of 5.3% (*p* < 0.001) and 8.8% (*p* < 0.001) over the same period.

### 3.10. Skin Youth Evaluation

Skin youthfulness was assessed by evaluating parameters associated with aging and photoaging, such as skin roughness. The CpLfe emulgel demonstrated a reduction in the appearance of wrinkles, with noticeable improvements in both expressive and gravitational wrinkles after 4 weeks of treatment. Specifically, the volume values of the forehead, frown lines, nasolabial folds, and crow’s feet were lower at both the 2- and 4-week assessments, as shown by their average percentage changes in Figure 4a,b.

In contrast, the placebo did not show any statistically significant improvements in skin roughness after 2 and 4 weeks of application.

Representative images of the forehead and frown lines, nasolabial folds, and crow’s feet of some panelists before (T_0_) and after the 4-week treatment (T_4w_) with the CpLfe emulgel or the placebo are shown in Figure 5a–d.

The CpLfe emulgel was found to significantly enhance collagen production in the dermis, thereby improving its structural organization. Collagen levels increased by 3.4% after 2 weeks (*p* < 0.01) and 8.5% after 4 weeks (*p* < 0.001). In contrast, the placebo demonstrated only minimal, non-significant changes in collagen production, with increases of 1.3% at 2 weeks (*p* > 0.05) and 2.1% at 4 weeks (*p* > 0.05). These changes relative to baseline (T_0_) are detailed in Table 9, and the most exemplary echography are shown in Figure 6.

Ultrasound skin imaging detects the acoustic response from the skin and subcutaneous tissues, where the reflected signal intensity is displayed on a color scale. Darker areas represent homogeneous composition, while brighter areas indicate changes in structural density. The images showed a notable increase in skin density in subjects treated with CpLfe, whereas the placebo had no significant effect on collagen levels.

### 3.11. Skin Lightening Activity Assessment

CpLfe treatment for 2 and 4 weeks also resulted in a statistically significant increase in the skin lightness, as measured by the L* parameter. Specifically, L* was enhanced by 1.0% and 2.2%, respectively (*p* < 0.05). In contrast to the placebo group, there was no significant variation in skin luminosity over time (Table 10).

Figure 7 displays comparative photos of some panelists before (T_0_) and after the 4-week treatment (T_4w_) with either the CpLfe emulgel or placebo. The photos are captured using cross-polarized light, which highlights the lightening benefits of the CpLfe over the placebo.

## 4. Discussion

### 4.1. Bioactive Compounds in CpL Flowers and Their Potential for Skin Health

The main objective of this investigation was to gather comprehensive data on the bioactive compounds present in the CpL flower in order to gain valuable insights into the potential use of this innovative source of bioactive compounds to support skin health and well-being. To achieve the goal, an investigation of the main molecular mechanism involved in skin cells in a pre-clinical model was carried out to assess the beneficial action of CpLfe and its suitability for topical application. Then a double-blind, placebo-controlled clinical study on 40 subjects was conducted to estimate Cplfe skin tolerability and activity.

The results highlight that CpL flowers represent a rich source of polyphenols and carotenoids, which are known to exert profitable efforts on skin health [71]. The UHPLC-Q-Orbitrap HRMS analysis allowed for a detailed analysis of the polyphenolic fraction detected in the flowers, providing quantitative data on the content of individual polyphenols. In spite of several investigations that have been carried out on the CpL flower, to our knowledge, the current work is the first study that investigates the polyphenolic profile of CpL flower through UHPLC-Q-Orbitrap HRMS analysis [4,50,51]. Overall, the main outcomes of the present study indicate that CpL flower may be a potential alternative source of polyphenolic compounds, such as rutin, isorhamnetin 3-rutinoside, kaempferol 3-glucoside, quercetin 3-galactoside, and kaempferol, as well as many other important polyphenols. Rutin appears as one of the most abundant polyphenols found in the tested CpLfe, accounting for 81% of the total polyphenol mixture. Rutin, known for its antioxidant and UV-protective properties [72], enhances sunscreen efficacy by boosting antioxidant activity by 40% and photoprotection by 70% when combined with UV filters, as confirmed by the unique in vivo studies that reported on its SPF-enhancing capabilities [73]. These effects help preserve the structural integrity of the skin by preventing the degradation of collagen and elastin, often caused by oxidative stress [74]. Additionally, rutin supports skin hydration by strengthening the skin barrier and reducing transepidermal water loss (TEWL), rather than directly improving water retention [75]. Studies, including one by Choi et al., have highlighted rutin’s ability to counteract skin aging caused by reactive oxygen species, which are a primary driver of intrinsic aging due to their continuous production during mitochondrial metabolism [76]. Furthermore, rutin anti-inflammatory effects, along with those of compounds found in the CpLfe, like isorhamnetin 3-rutinoside, quercetin, and kaempferol, help reduce skin irritation, promote healing [77], and enhance collagen synthesis, further supporting skin elasticity and reducing wrinkles.

The polyphenol composition of CpL flowers was previously analyzed by Mohamed et al. [78] through HPLC-DAD. Their findings indicated that the predominant polyphenols in the methanolic extract of the CpL flowers were isorhamnetin, quercetin, and myricetin. These polyphenols, known for their antioxidant properties [74,79,80], likely play a role in reducing oxidative stress in the skin, which can help maintain the skin’s structure and function, improving hydration [79]. Morittu et al. [4] also investigated the polyphenol composition of CpL flowers using HPLC-DAD analysis and detected rutin, syringic acid, catechin, epicatechin, and hesperidin as the main components in the ethanolic extract. Moreover, the TPC data found in the assayed extracts of CpL flowers were lower than those reported by Loubet-González [81], who reported a TPC value of 876 mg GAE/100 g DW for methanol extracts of CpL flowers obtained with the Soxhlet procedure. These differences may be related to the different extraction procedures and/or the different solvents used. Moreover, our findings revealed a two-fold increase in the TPC value than the data previously reported by Aquino-Bolaños et al. [46] for ethanol extracts of the flowers.

On the other hand, our data showed a high content of carotenoids in CpL flowers, particularly lycopene. These findings align with the results of Majumder et al. [82], who also found a high presence of lycopene in the same flower species. Carotenoids have significant antioxidant properties that can positively impact skin wellness. These natural compounds are believed to provide a range of benefits, including protection against sun damage, enhanced skin hydration, and a reduction in the appearance of fine lines and wrinkles [83,84,85,86]. Lycopene, in particular, has been linked to enhanced skin hydration [87], likely through its ability to support the skin’s natural barrier function, thus preventing transepidermal water loss (TEWL). Furthermore, carotenoids such as lycopene are believed to reduce the appearance of fine lines and wrinkles by preventing the breakdown of collagen and elastin fibers in the skin [88]. These outcomes highlighted the potential of CpL flowers as a source of biologically active compounds that can contribute to overall skin health, youthfulness, and well-being, which was further predicted by in vitro antioxidant and cellular essays and verified with the clinical trial.

CpLfe showed noticeable antioxidant activity also in a cellular pre-clinical model by reducing intercellular ROS and exerting a protective activity against UVB irradiation. In fact, the MTT assay performed on HaCaT cells with different concentrations of CpLfe (100, 250, and 500 μg/mL) and irradiated with UVB at 30 mJ/cm^2^ revealed that CpLfe pre-treatment showed a dose-dependent protective effect, with cell viability at 84% even at the lowest dose. At 250 and 500 μg/mL, cell viability was almost comparable to non-irradiated controls. The effects of UVB treatment were further evaluated by examining the expression levels of MMP-1, a marker associated with photoaging. UVB irradiation at 30 mJ/cm^2^ increased MMP-1 expression by about 30%. Pre-treatment with CpLfe significantly reduced MMP-1 levels, especially at 250 and 500 μg/mL doses. Therefore, as established in the human HaCaT cell line, these findings suggested that CpLfe can effectively mitigate UVB-induced molecular damage, highlighting its potential as a protective agent against photoaging. Furthermore, the reduction in MMP-1 expression indicates that CpLfe might prevent the degradation of extracellular matrix components, thereby maintaining skin integrity and function. These findings were proven in the clinical trial, where ultrasound sonography analysis indicated higher collagen levels in the dermis (8.5% at T_4w_, *p* < 0.001) in panelists treated with CpLfe, with significant differences compare to the placebo (*p* < 0.05). Still, as known, UVB exposure leads to ROS overproduction that can be detrimental, leading to the degradation of several cellular components, like membranes or proteins. The protective action of CpLfe together with its ability to stimulate ceramide production indicates that it could be a valuable component in products aimed at improving skin barrier integrity. We found that mRNA expression levels of CerS2, CerS4, and CerS6 significantly increased in cells treated with CpLfe (250 μg/mL), suggesting that CpLfe may enhance the expression of ceramide synthases, which are crucial for ceramide production and thus for maintaining and restoring the human skin barrier function [36,89,90,91,92]. According to this, the clinical trial evidenced a reduced trans epidermal water loss (TEWL) in subjects treated with CpLfe (7.5% at T_2w_; 10.2% at T_4w_, *p* < 0.05), indicating the ability of the extract to recover the skin barrier integrity and prevent excessive dehydration [75,93]. To estimate the hydration properties of CpLfe, we measured AQP3 mRNA levels using real-time PCR analysis. After 24 h of exposure to a 250 µg/mL dose, CpLfe-treated cells exhibited significantly increased AQP3 levels compared to untreated and vehicle controls. This suggests that CpLfe enhances skin hydration by upregulating AQP3 expression. At the same time, CpLfe demonstrated the ability to also increase skin hydration even after two weeks of treatment, with significantly increased corneometry values (13.7% at T_2w_; 15.0% at T_4w_, *p* < 0.05).

Regarding tolerability, the present study showed that no adverse effects were observed after repeated application of CpLfe emulgel for 4 weeks. Our findings suggest that CpLfe may be an effective skin-protecting agent, with applications in the treatment of skin dermal laxity and wrinkles, ultimately promoting skin wellness.

### 4.2. Research Limitations and Strengths

The key strength of this research is its thorough investigation of Cucurbita pepo L. flower extract (CpLfe), demonstrating significant antioxidant, photoprotective, and hydration-boosting effects, validated through in vitro assays and a placebo-controlled clinical trial. Nonetheless, some limitations should be noted. The study’s focus on controlled experimental conditions may not fully capture its performance in real-world cosmetic applications. Additionally, while CpLfe effectively improved hydration and reduced oxidative stress, its long-term effects on skin aging and its stability in diverse formulations remain unexplored. Future research should address these aspects to establish CpLfe’s broader applicability in skincare.

## 5. Conclusions

Skin homeostasis is also crucially dependent on the expression and presence of aquaporins. These membrane proteins facilitate water transport across cells, ensuring optimal hydration and overall skin health. Accumulating evidence indicates that the water-, glycerol-, and hydrogen peroxide-transporting channel aquaporin-3 (AQP3) plays a key role in keratinocyte function, with abnormalities observed in several human skin diseases [94]. Polyphenols, such as rutin, present in high concentrations in CpLfe, have been suggested to modulate aquaporin expression through their antioxidant and signaling properties by promoting the upregulation of AQP3.

This study aimed to evaluate the bioactive compounds in CpL flowers and their potential benefits for skin health. The findings revealed that CpL flowers are rich in polyphenols and carotenoids, which possess significant antioxidant properties. Pre-clinical models showed that CpL flower extract (CpLfe) effectively reduces intracellular ROS, protects against UVB damage, and enhances ceramide synthase expression, contributing to skin barrier integrity. Clinical trials confirmed CpLfe’s ability to increase skin hydration, reduce trans epidermal water loss, and boost collagen levels without adverse effects. These results suggest that CpLfe is a promising ingredient for promoting skin wellness and protection.

## Figures and Tables

**Figure 1 antioxidants-13-01476-f001:**
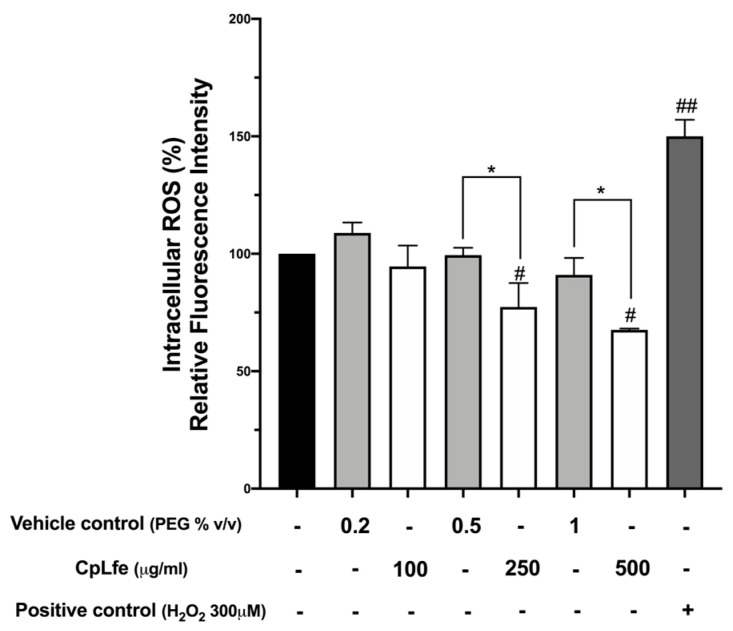
Evaluation of intracellular ROS level in HaCaT cells. The effect of CpLfe at 100, 250, and 500 μg/mL on intracellular ROS levels evaluated using the H2DCF-DA assay after 24 h treatment as compared with the vehicle control. Cells treated only with H_2_O_2_ (300 μM) were used as a positive control. The graph represented the mean and SD of three separate experiments and relative fluorescence intensity was calculated as fold-change relative to untreated control cells, arbitrarily set at 100%. Differences versus each respective negative control (mock) were considered significant at * *p*-value ≤ 0.05, and versus untreated cells at # *p*-value ≤ 0.05, ## *p*-value ≤ 0.0001 (calculated by one-way ANOVA, followed by Dunnett’s multiple comparisons test).

**Figure 2 antioxidants-13-01476-f002:**
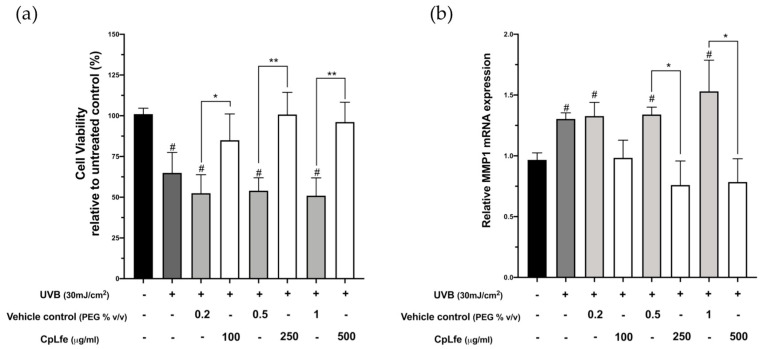
Effect of CpLfe on UVB-induced damage in HaCaT cells. Cells were pre-treated with the indicated concentrations of CpLfe compound for 24 h, irradiated with UVB (30 mJ/cm^2^), and incubated for 24 h. Negative controls were included in each experiment by treating cells with the appropriate amount of polyethylene glycol (vehicle). (**a**) Cell viability was analyzed by MTT assay after UVB exposure. (**b**) MMP-1 RNA levels were analyzed using real-time PCR as described in Materials and Methods. Data represented the mean ± SEM from baseline of three independent experiments. Differences versus each respective negative control (vehicle) were considered significant at * *p*-value ≤ 0.05 and ** *p*-value ≤ 0.0001 and versus untreated cells at # *p*-value ≤ 0.05 and (calculated by one-way ANOVA, followed by Dunnett’s multiple comparisons test).

**Figure 3 antioxidants-13-01476-f003:**
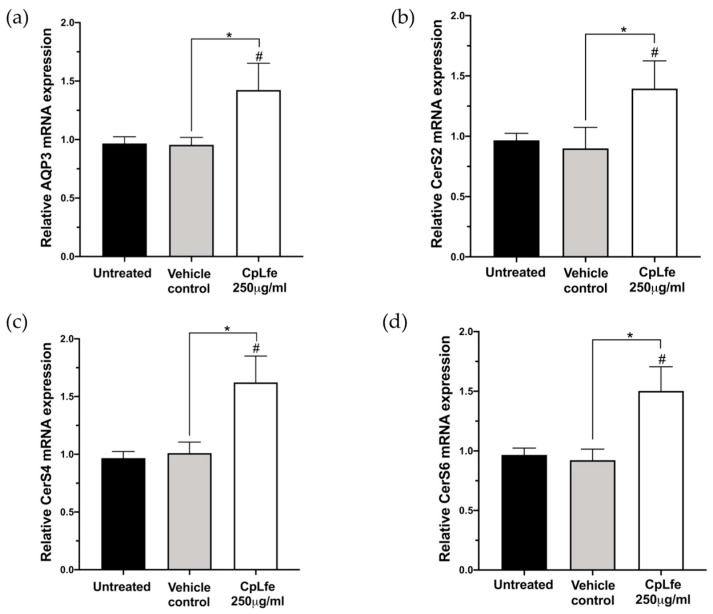
Effect of CpLfe on the mRNA expression level of aquaporin-3 (AQP-3), ceramide synthases CerS2, CerS4, and CerS6 in HaCaT cells. Quantitative real-time PCR analysis showed increased mRNA expression levels for AQP-3 (**a**), CerS2 (**b**), CerS4 (**c**), and CerS6 (**d**) in HaCaT cells after CpLfe treatment. Expression levels were normalized to those of glyceraldehyde-3-phosphate dehydrogenase (GAPDH) used as housekeeping control. Data represented the mean ± SEM from baseline of three independent experiments. Differences versus vehicle control (polyethylene glycol 0.5%, *v*/*v*) were considered significant at * *p*-value ≤ 0.05 and versus untreated cells at # *p*-value ≤ 0.05 (calculated by one-way ANOVA, followed by Dunnett’s multiple comparisons test).

**Figure 4 antioxidants-13-01476-f004:**
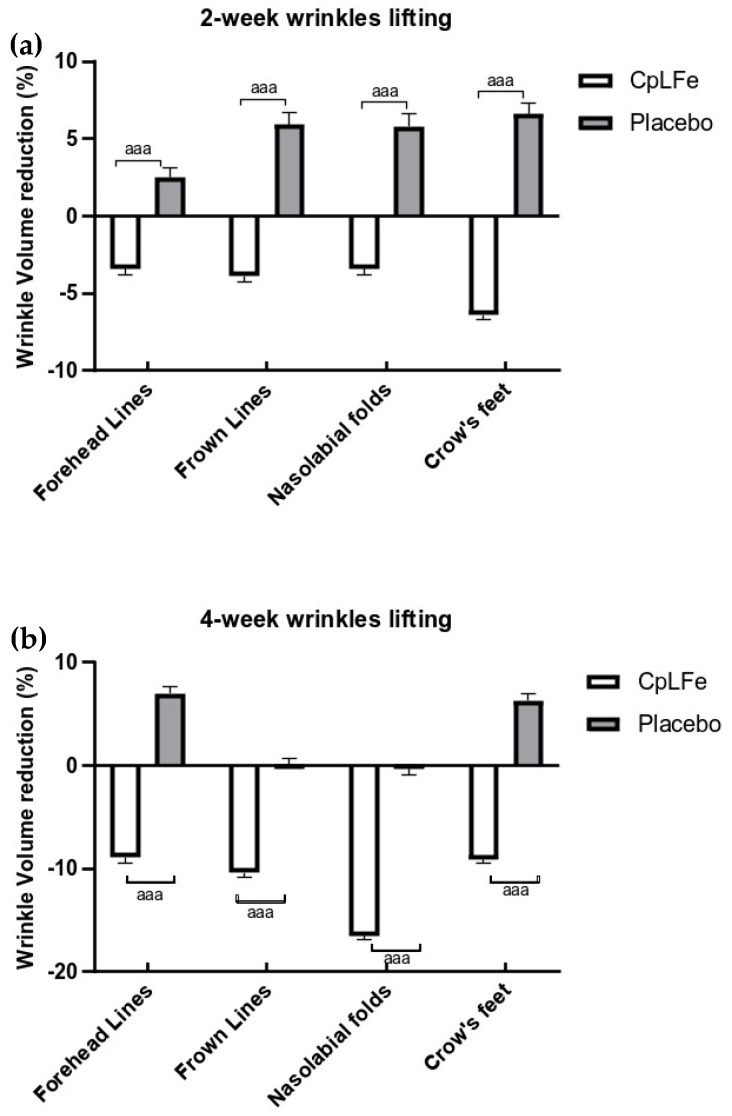
Response rate (%) ± SEM from baseline on skin wrinkles at two efficacy endpoints. (**a**) 2-week results, (**b**) 4-week results. All the variations between groups are statistically significant at both check-ups (ANOVA test, ^aaa^
*p* < 0.001).

**Figure 5 antioxidants-13-01476-f005:**
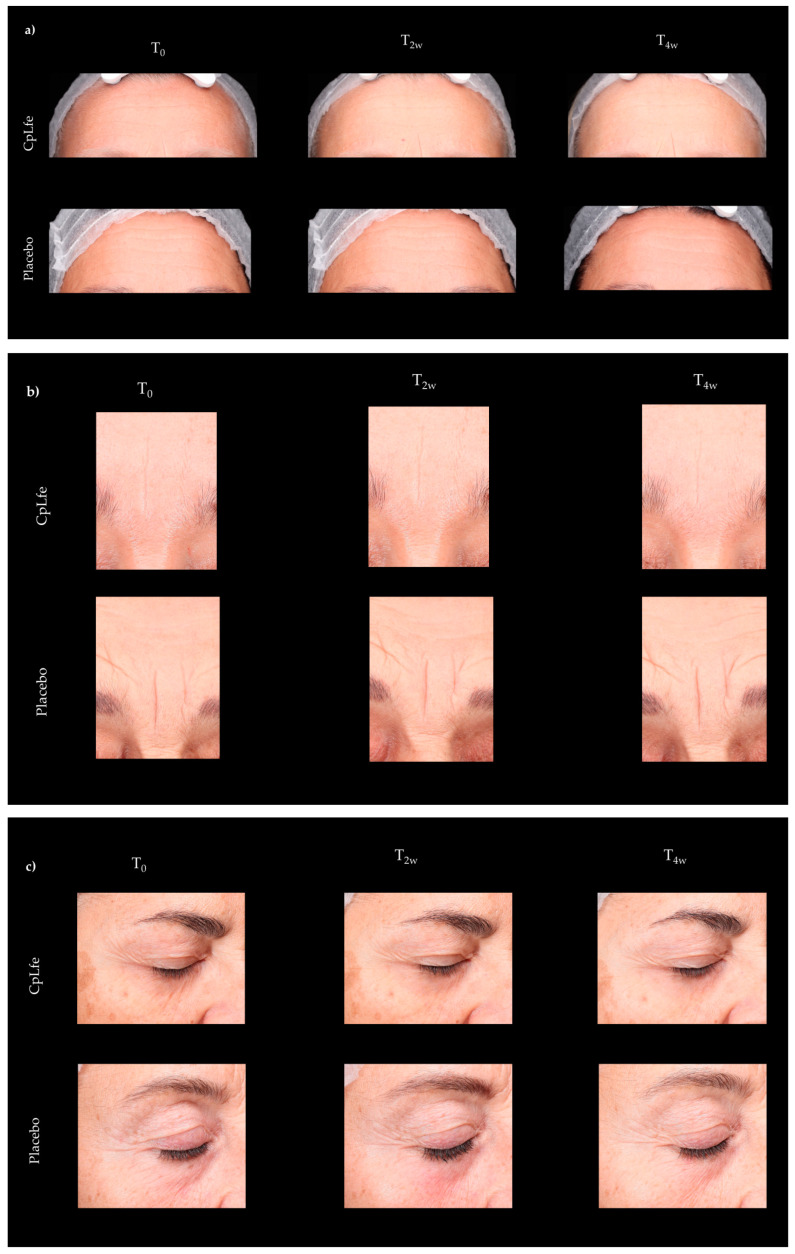
Representative photographs taken with using VISIA 7th (Canfield Scientific Inc., Parispanny, NY, USA) of (**a**) forehead wrinkles, (**b**) frown lines, (**c**) crow’s feet, and (**d**) nasolabial folds of subjects treated with CpLfe emulgel or a placebo for 4 weeks.

**Figure 6 antioxidants-13-01476-f006:**
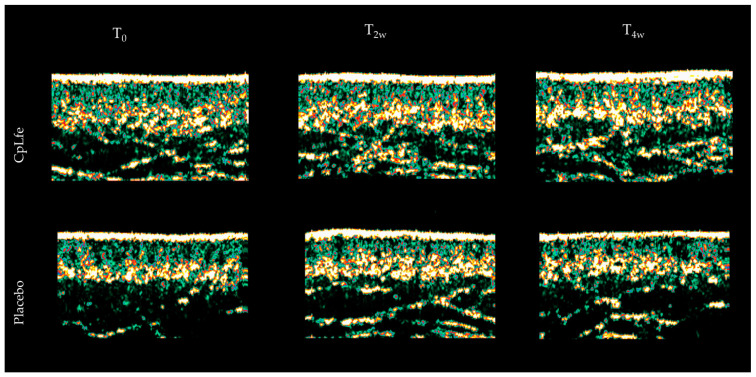
Illustrative facial echography taken with Dermascan C 20 MHz of subjects treated with CpLfe emulgel or placebo for 4 weeks.

**Figure 7 antioxidants-13-01476-f007:**
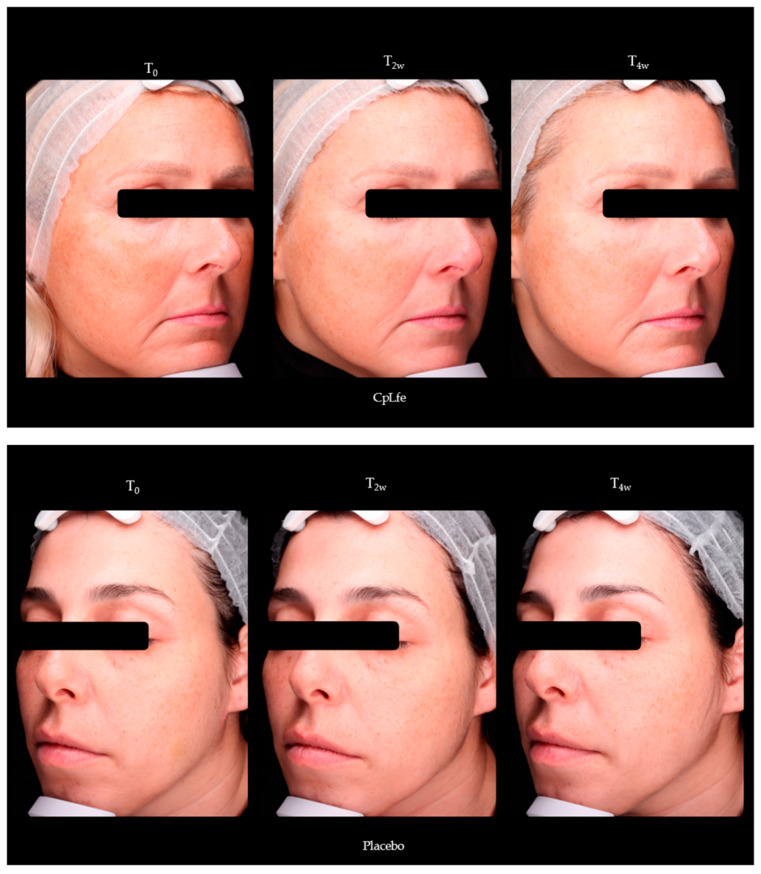
Representative photographs of subjects treated with CpLfe or placebo for 4 weeks. Photos are captured using VISIA 7th with cross-polarized light to appreciate the overall complexion.

**Table 1 antioxidants-13-01476-t001:** Primer sequences used for quantitative real-time PCR analysis.

Transcript (Accession Number)	Primer	Sequence 5′-3′	Amplicon Size (bp)
AQP3 (NM_004925.5)	For	AGATGCTCCACATCCGCTAC	143
	Rev	GGTTGATGGTGAGGAAACCA	
CERS2 (NM_022075.5)	For	CCGATTACCTGCTGGAGTCAG	121
	Rev	GAAGGGCAGGATGACCAGTC	
CERS4 (NM_024552.3)	For	CTTCGTGGCGGTCATCCTG	77
	Rev	TGTAACAGCAGCACCAGAGAG	
CERS6 (NM_001256126.2)	For	GGGATCTTAGCCTGGTTCTGG	183
	Rev	CGCACGGTTTGGCTACAAATC	
MMP-1 (NM_002421.4)	For	TGCGTGCGCACAAATCCCTTCTAC	79
	Rev	TTCAAGCCCATTTGGCAGTT	
GAPDH (NM_002046.7)	For	GAGCCACATCGCTCAGACAC	116
	Rev	GGCAACAATATCCACTTTACCA	

**Table 2 antioxidants-13-01476-t002:** Qualitative and quantitative composition of the investigated emulgels.

CpLfe			Placebo		
Phase	Ingredients (INCI)	Quantity (%)	Phase	Ingredients (INCI)	Quantity (%)
W	Aqua	qs to 100	W	Aqua	qs to 100
	Sodium Gluconate	0.2		Sodium Gluconate	0.2
	Glycerin	3.0		Glycerin	3.0
	Carbomer	0.3		Carbomer	0.3
O	Sodium Polyacrylate (and) Dicaprylyl Carbonate (and) Polyglyceryl-3 Caprate	1.5	O	Sodium Polyacrylate (and) Dicaprylyl Carbonate (and) Polyglyceryl-3 Caprate	1.5
	Caprylic/Capril Triglycerides	5.0		Caprylic/Capril Triglycerides	5.0
	Coconut Alkanes (and) Coco-Caprylate Caprate	7.0		Coconut Alkanes (and) Coco-Caprylate Caprate	7.0
	Octil 2-Dodecanol	5.0		Octil 2-Dodecanol	5.0
C	Phenoxyethanol and Ethylhexylglycerin	0.9	C	Phenoxyethanol and Ethylhexylglycerin	0.9
	CpLfe 5%_w/v_	0.5		NaOH sol. 20%	0.1
	NaOH sol. 20%	0.1			

qs: quantity sufficient.

**Table 3 antioxidants-13-01476-t003:** UHPLC-Q-Orbitrap HRMS parameters of the studied analytes (n = 24).

Analytes	Adduct Ion	Chemical Formula	RT (min)	Theoretical Mass (*m*/*z*)	Measured Mass (*m*/*z*)	Accuracy (Δ ppm)	LOD (mg/kg)	LOQ (mg/kg)
Quinic acid	[M-H]^−^	C_7_H_12_O_6_	0.47	191.05531	191.05611	4.19	0.026	0.078
Protocatechuic acid	[M-H]^−^	C_7_H_6_O_4_	2.31	153.01930	153.01857	−4.77	0.013	0.039
Clorogenic Acid	[M-H]^−^	C_16_H_18_O_9_	3.00	353.08780	353.08798	0.51	0.013	0.039
Epicatechin	[M-H]^−^	C_15_H_14_O_7_	3.17	289.07176	289.07202	0.90	0.013	0.039
Caffeic acid	[M-H]^−^	C_9_H_8_O_4_	3.23	179.03498	179.03455	−2.40	0.013	0.039
Catechin	[M-H]^−^	C_15_H_14_O_6_	3.34	289.07175	289.07205	1.04	0.026	0.078
*p*-Coumaric acid	[M-H]^−^	C_9_H_8_O_3_	3.46	163.04001	163.03937	−3.92	0.013	0.039
Vitexin	[M-H]^−^	C_21_H_20_O_10_	3.48	431.09837	431.09711	−2.92	0.013	0.039
Apigenin-7-O-glucoside	[M-H]^−^	C_15_H_10_O_5_	3.49	269.04555	269.04526	−1.08	0.026	0.078
Ferulic acid	[M-H]^−^	C_10_H_10_O_4_	3.55	193.05063	193.05016	−2.43	0.026	0.078
Naringin	[M-H]^−^	C_27_H_32_O_14_	3.56	579.17193	579.17212	0.33	0.013	0.039
Quercetin 3 galactoside	[M-H]^−^	C_21_H_20_O_12_	3.58	463.08820	463.08817	−0.06	0.026	0.078
Rutin	[M-H]^−^	C_27_H_30_O_16_	3.59	609.14611	609.14673	1.02	0.013	0.039
Diosmin	[M-H]^−^	C_28_H_31_O_15_	3.64	607.16684	607.16534	−2.47	0.013	0.039
Kaempferol 3-glucoside	[M-H]^−^	C_21_H_20_O_11_	3.68	447.09195	447.09329	3.00	0.013	0.039
Isorhamnetin 3-rutinoside	[M-H]^−^	C_28_H_32_O_16_	3.72	623.16176	623.16174	−0.03	0.013	0.039
Miricetin	[M-H]^−^	C_14_H_10_O_8_	3.73	317.03029	317.02924	−3.31	0.013	0.039
Daidzein	[M-H]^−^	C_15_10_0_O_4_	3.77	253.05063	253.05035	−1.11	0.026	0.078
Quercetin	[M-H]^−^	C_15_H_10_O_7_	3.88	301.03538	301.03508	−1.00	0.013	0.039
Naringenin	[M-H]^−^	C_15_H_12_O_5_	3.91	271.06120	271.06110	−0.37	0.013	0.039
Luteolin	[M-H]^−^	C_15_H_10_O_6_	3.98	285.04046	285.04086	1.40	0.026	0.078
Kaempferol	[M-H]^−^	C_15_H_10_O_6_	4.01	285.04046	285.04086	1.41	0.013	0.039
Genistein	[M-H]^−^	C_15_H_10_O_5_	4.05	269.04554	269.04562	0.30	0.013	0.039
Apigenin	[M-H]^−^	C_15_H_10_O_5_	4.08	269.04555	269.04556	0.04	0.026	0.078

RT: retention time; LOD: limit of detection; LOQ: limit of quantification.

**Table 4 antioxidants-13-01476-t004:** Polyphenol content in *Cucurbita pepo* L. flower.

Analyte	Average (mg/kg)	±SD
*p*-Coumaric acid	11.68	0.11
Naringin	<LOQ	
Quercetin 3-galactoside	26.9	0.71
Rutin	2359.5	0.28
Kaempferol 3-glucoside	6.58	0.09
Isorhamnetin 3-rutinoside	497.8	2.55
Myricetin	2.78	0.04
Quercetin	0.52	0.06
Kaempferol	0.82	0.09
Apigenin	<LOQ	

LOQ: limit of quantification.

**Table 5 antioxidants-13-01476-t005:** Carotenoids content in *Cucurbita pepo* L. flower. The results are expressed as mg/kg of the dry weight.

Sample	Lutein		β-Carotene		Lycopene	
	Mean	±SD	Mean	±SD	Mean	±SD
*Cucurbita pepo* L. flower	407	32	83	12	513	41

**Table 6 antioxidants-13-01476-t006:** Antioxidant activity and total phenolic compounds of *Cucurbita pepo* L. flower.

Sample	DPPH		ABTS		FRAP		TPC	
	Mean	±SD	Mean	±SD	Mean	±SD	Mean	±SD
*Cucurbita pepo* L. flower extract	10.7	0.2	12.4	0.4	77.5	0.6	534.2	12.9

TPC: total polyphenolic content; antioxidant activity assays: DPPH: 2,2-diphenyl-1-picrylhydrazyl; ABTS: 2,2′-azino-bis(3-ethylbenzothiazoline-6-sulfonic acid); FRAP: ferric-reducing antioxidant power.

**Table 7 antioxidants-13-01476-t007:** Measure of pH values and viscosity of the investigated emulgel: CpLfe and placebo.

Sample	pH		mPa (L4, 20 rpm)	
	After Preparation	After 24 h	After Preparation	After 24 h
CpLfe	5.3	5.5	21.254	22.156
Placebo	5.2	5.4	21.136	21.723

**Table 8 antioxidants-13-01476-t008:** Skin wellness parameters change from baseline to week 4 for the two studied groups. (*t*-test Student vs. baseline *** *p* < 0.001).

		Baseline (T_0_) Mean ± SD	Week 2 (T_2w_) Mean ± SD	∆% from Baseline	Week 4 (T_4w_) Mean ± SD	∆% from Baseline
CpLfe	TEWL (g/h/m^2^)	6.5 ± 0.4	6.0 ± 0.4	−7.5% ***	5.8 ± 0.2	−10.2% ***
	Hydration (U.A.)	65.6 ± 4.0	74.3 ± 1.3	13.7% ***	75.2 ± 2.4	15.0% ***
Placebo	TEWL (g/h/m^2^)	6.4 ± 0.8	6.7 ± 0.6	6.4%	6.5 ± 0.3	2.9%
	Hydration (U.A.)	55.7 ± 2.2	58.6 ± 1.2	5.3% ***	60.5 ± 1.2	8.8% ***

**Table 9 antioxidants-13-01476-t009:** Collagen index changes at the dermis layer from baseline to week 4 for the two studied groups. (*t*-test Student vs. baseline ** *p* < 0.01, *** *p* < 0.001).

	Collagen Index			
	CpLfe		Placebo	
Timing	Average ± SD	Average ∆% vs. Baseline	Average ± SD	Average ∆% vs. Baseline
Baseline	44.2 ± 1.6		47.2 ± 1.9	
After 2-week treatment (T_2w_)	45.6 ± 0.8	3.4% **	47.8 ± 2.4	1.3%
After 4-week treatment (T_4w_)	47.9 ± 0.9	8.5% ***	48.3 ± 2.3	2.1%

All differences between groups were statistically significant *p* < 0.05 (ANOVA test).

**Table 10 antioxidants-13-01476-t010:** Skin lightness changes on panelists skin from baseline to week 4 for the two studied groups–Comparison before/after on dark area. (*t*-test Student vs. baseline ** *p* < 0.01, *** *p* < 0.001).

	L*			
	CpLfe		Placebo	
Timing	Average ± SD	Average ∆% vs. Baseline	Average ± SD	Average ∆% vs. Baseline
Baseline	63.1 ± 0.7		62.5 ± 0.9	
After 2-week treatment (T_2w_)	63.8 ± 0.8	1.0% **	62.4 ± 0.8	0.00%
After 4-week treatment (T_4w_)	64.5 ± 0.7	2.2% ***	62.8 ± 0.7	0.60%

All differences between groups were statistically significant at T_4w_, *p* < 0.05 (ANOVA test). L* luminosity.

## Data Availability

Data are contained within the article and Appendix A.

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
