# Peer review of "Chemical Profile and Promising Applications of Cucurbita pepo L. Flowers"

_antioxidants, 2024, doi:10.3390/antiox13121476_

Round 1
Reviewer 1 Report (Previous Reviewer 3)
Dear Authors,
For further manuscript evaluation and peer-review process, please provide the approval of the study by the local ethics committee. Even being a cosmetic protocol, the participants require the attention of the local ethics committee.
Please provide the approval of the study by the local ethics committee
Author Response
Response:
On behalf of all the authors, I thank you for evaluating our manuscript. We understand your concern about the ethical considerations related to human testing. Please be assured that we carefully addressed these ethical concerns before conducting our study.
Specifically, we conducted an occlusive patch test for 48 hours, prepared a detailed Material Safety Data Sheet (MSDS), and provided all enrolled participants with the complete ingredient list of the materials used in the cosmetic formula. Nevertheless, we appreciate the additional insights provided and have taken prompt action to secure further documentation.
We have obtained written approval from the Institutional Review Board of the Department of Pharmacy at the University of Naples, an independent body that reviews ethical compliance for studies involving non-drug, minimally invasive methods. This approval, along with the Safety Assessment 48-Hour Patch Test and the MSDS, is attached to this email for your reference. Additionally, we have registered the clinical trial on ClinicalTrials.gov under the identifier NCT06674005 (https://clinicaltrials.gov/study/NCT06674005).
We hope this addresses the ethical concerns raised and demonstrates our commitment to meeting the highest ethical standards. Please let us know if there are any additional concerns you would like us to take.

Reviewer 2 Report (Previous Reviewer 4)
The revised paper has been well revised and I agree to accept it for publication.
The revised paper has been well revised and I agree to accept it for publication.
Author Response
Thank you for evaluating our manuscript
Round 2
Reviewer 1 Report (Previous Reviewer 3)
Dear Authors,
Thank you for the opportunity to read your research work.
I write you in regard to your manuscript titled "Chemical Profile and Promising Applications of Cucurbita pepo L. Flowers".
Your investigation can be considered robust with interesting results. Overall, the entire manuscript must be revised, reorganized, and more discussed.
- please, revise Table 2. For instance, remove the plural of the word "emulgels" or add the blank sample.
- Table 3 should be in the results section.
- Tables 4, 5, and 6 require legend.
- please, revise if the results section has mirrored the material and methods section.
- Table 8 lacked units.
- please, describe in material and methods section the procedures of the results in item 3.8.
- considering the results, several methods were superficially or not described. Please, explain all the techniques that justify a result. How was the collagen index obtained?
- please, add a paragraph about limitations and strengths.
- please, revise the discussion section associating the clinical outcomes with the composition of the natural ingredient. for instance, rutin has been investigated as a sunscreen booster and its ability to improve skin hydration was not observed.
Author Response
Comments
- please, revise Table 2. For instance, remove the plural of the word "emulgels" or add the blank sample.
In line 309, the table was revised to include the INCI of both the formulation containing the active ingredient and the placebo formulation.
- Table 3 should be in the results section.
Table 3 has been moved to the Results section at line 500, becoming Table 7. It is now included in the new paragraph 3.8 titled "Physicochemical Characterization of Formulations" at line 497.
- Tables 4, 5, and 6 require legend.
Legend has been added to the indicated tables
- please, revise if the results section has mirrored the material and methods section.
The authors have rearranged the paragraphs according to the reviewer's indications.
- Table 8 lacked units.
Units of measurement have been added to Table 8 at line 509.
- please, describe in material and methods section the procedures of the results in item 3.8.
Paragraph 2.9.4, "Skin Condition Analysis," in the Materials and Methods section, has been revised to include the procedures used to measure skin parameters.
- considering the results, several methods were superficially or not described. Please, explain all the techniques that justify a result. How was the collagen index obtained?
Paragraphs 3.9, 3.10, and 3.11, which present the clinical trial results, have been revised to better clarify how the outcomes were obtained.
- please, add a paragraph about limitations and strengths.
In line 655, paragraph 4.2 has been added, discussing the limitations and strengths of the research.
- please, revise the discussion section associating the clinical outcomes with the composition of the natural ingredient. for instance, rutin has been investigated as a sunscreen booster and its ability to improve skin hydration was not observed.
Based on comments, the authors revised the discussion section.
Round 3
Reviewer 1 Report (Previous Reviewer 3)
Authors have improved their manuscript.
Please, to avoid misinterpretation with tissue or human tests, associate your results from the UVB-derived assay with the cell culture, for example, in line 668: the photoprotective effect was established in a cell culture, suggesting this type of efficacy.
Author Response
Please consider consulting more cosmetic attributes from Rutin, including the trial for SPF, the only assay that proves photoprotection efficacy.
We appreciate your suggestion and would like to clarify that we have extensively explored the cosmetic attributes of rutin, as detailed in lines 591–605. Furthermore, we have incorporated the trial you suggested to enhance the quality and robustness of our assertions regarding rutin's cosmetic benefits.
Please, to avoid misinterpretation with tissue or human tests, associate your results from the UVB-derived assay with the cell culture, for example, in line 668: the photoprotective effect was established in a cell culture, suggesting this type of efficacy.
As suggested, we have clarified the sentence in line 668.
Thank you for your valuable comment and contribution to improving our work.
This manuscript is a resubmission of an earlier submission. The following is a list of the peer review reports and author responses from that submission.
Round 1
Reviewer 1 Report
Comments and Suggestions for Authors
From the author's response, it is still hard to find the novelty of this work.
(1) Although this work has determined the antioxidant activity of the flower; how its antioxidant property contributes to its anti-aging activities, this part did not investigate. Thus, this study lacks of significance and novelty.
(2) Both positive and negative modes should be used to screen the potential bioactive compounds. Some bioactive compounds may be missed if only the negative mode is used.
(3) Although this study found that the flower exerted anti-aging effects, there are no detailed mechanisms are discussed, particularly the antioxidant activity involved in anti-aging
Therefore, this study is not suitable for publication in Antioxidants.
Comments on the Quality of English LanguageModerate editing of English language required
Reviewer 2 Report
Comments and Suggestions for Authors
The aim of this paper was to investigate the impact of incorporating Cucurbita pepo L. flower extract into an emulgel cosmetic formulation on skin barrier recovery, hydration, collagen levels, wrinkles, and melasma lesions. The author’s work on discussing achieved results is appreciated. The title is clear and adequate to the article’s content. The authors have responded adequately to my previous comments.
Comments on the Quality of English LanguageMinor editing of English language required
Reviewer 3 Report
Comments and Suggestions for Authors
Dear Authors,
I write you in regard to your manuscript "Chemical Profile and Promising Skin Applications of Cucurbita pepo L. Flowers".
- please, in line 92, the controlled released profile could be a case by case matter.
- still in line 92, for a topical product, absorption must be avoided. Please, correct the sentence. a review of this pharmaceutical form must be performed.
- considering the in vivo test involving humans, there must be a previous approval of the investigation by a local ethics committee.
I believe, from this point, the peer review process should continue after the presentation of the ethics committee approval.
Reviewer 4 Report
Comments and Suggestions for Authors
This study demonstrated CpLfe's antioxidant efficacy, component analysis, clinical anti-wrinkle efficacy, and pigmentation improvement efficacy. Minor revisions are needed for the paper to be published.
1) Table 7; Add the p value explanation.
2) Table 8, 9, 10, 11; When notating p value, indicate only the p value that appears in the table. For example, in table 7 legend, *p<0.05 is not necessary.
3) The excellent antioxidant effect of CpLfe was verified through in vitro assay. It is well known that the antioxidant effect affects collagen increase, MMP inhibition, and pigmentation inhibition, and the author explained this and showed it as a clinical effect. However, in order to more clearly identify the mechanism, it would be better if data showing tyrosinase activity inhibition assay or MMP inhibition assay using CpLfe were added, so it is necessary to add data or mention this in the discussion.